# The cellular basis of mechanosensory Merkel-cell innervation during development

Blair A Jenkins[1,2], Natalia M Fontecilla[1], Catherine P Lu[3†], Elaine Fuchs[3], Ellen A Lumpkin[1]*

[1]Department of Physiology and Cellular Biophysics, Columbia University, New York, United States; [2]Department of Dermatology, Columbia University, New York, United States; [3]Robin Neustein Laboratory of Mammalian Development and Cell Biology, Howard Hughes Medical Institute, The Rockefeller University, New York, United States

**Abstract** Touch sensation is initiated by mechanosensory neurons that innervate distinct skin structures; however, little is known about how these neurons are patterned during mammalian skin development. We explored the cellular basis of touch-receptor patterning in mouse touch domes, which contain mechanosensory Merkel cell-neurite complexes and abut primary hair follicles. At embryonic stage 16.5 (E16.5), touch domes emerge as patches of Merkel cells and keratinocytes clustered with a previously unsuspected population of *Bmp4*-expressing dermal cells. Epidermal Noggin overexpression at E14.5 disrupted touch-dome formation but not hair-follicle specification, demonstrating a temporally distinct requirement for BMP signaling in placode-derived structures. Surprisingly, two neuronal populations preferentially targeted touch domes during development but only one persisted in mature touch domes. Finally, Keratin-17-expressing keratinocytes but not Merkel cells were necessary to establish innervation patterns during development. These findings identify key cell types and signaling pathways required for targeting Merkel-cell afferents to discrete mechanosensory compartments.

DOI: https://doi.org/10.7554/eLife.42633.001

*For correspondence:
ellen.a.lumpkin@gmail.com

Present address: †Departments of Plastic Surgery and Cell Biology, New York University, New York, United States

## Introduction

Touch, our most intimate sense, requires direct contact between skin and objects in our environment. In newborn altricial animals, touch is a key sensory modality through which parents nurture offspring so that they thrive. For example, neonatal rodent pups respond vigorously to their mother's nurturing licks and use tactile cues to learn to feed (*Braun and Champagne, 2014*). In the absence of caregiver touch, mammals display impaired cognitive development and social behaviors that can persist into adulthood (*Blakemore et al., 2006*; *Cascio et al., 2008*; *Orefice et al., 2016*; *Tomchek and Dunn, 2007*). To understand how touch and cognitive development are linked, a deeper understanding of the mechanisms underlying mammalian touch receptor formation is needed.

Precise patterning of different types of touch receptors enables proper relay of touch modalities, such as brushing, stroking and gentle pressure, from skin to the central nervous system (*Adrian and Zotterman, 1926a*; *Adrian and Zotterman, 1926b*; *Bai et al., 2015*; *Brown and Iggo, 1967*; *Burgess et al., 1968*; *Cheng et al., 2018*; *Iggo and Muir, 1969*; *Li and Ginty, 2014*; *Li et al., 2011*; *Rutlin et al., 2014*; *Zotterman, 1939*). For example, Merkel cell-neurite complexes, which are found in sensitive skin areas such as fingertips, are epithelial-derived cells that complex with myelinated sensory afferents to encode object features and gentle pressure (*Iggo and Muir, 1969*;

*Johnson, 2001*; *Maksimovic et al., 2014*; *Maricich et al., 2012*; *Wellnitz et al., 2010*; *Woodbury and Koerber, 2007*). In hair-bearing skin, Merkel cell-neurite complexes reside in touch domes and some hair follicles such as vibrissae. Recent work has revealed molecular pathways through which epidermal progenitors adopt a Merkel-cell fate (*Morrison et al., 2009*; *Nguyen et al., 2018*; *Perdigoto et al., 2013*; *Perdigoto et al., 2014*; *Van Keymeulen et al., 2009*; *Xiao et al., 2016*); however, little is known about how Merkel cell-containing mechanosensory structures form during mammalian skin development.

In vertebrates, many sensory epithelia derive from placodes. These thickened patches of epithelial cells engage in molecular crosstalk with mesenchymal cells to form specialized tissues that are selectively innervated by sensory neurons (*Hall et al., 1999*; *Whitfield, 2015*). Placode-derived sensory cells include taste cells and mechanosensory hair cells of the acoustico-lateralis system (*Agarwala et al., 2015*; *Dambly-Chaudière et al., 2003*; *Metcalfe et al., 1985*). Like taste cells, Merkel cells are Keratin 8 (K8) expressing cells that are derived from the Keratin 14 lineage (*Morrison et al., 2009*; *Van Keymeulen et al., 2009*). Like hair cells, Merkel cells are mechanosensory receptor cells that rely on the proneural transcription factor Atoh1 for specification (*Morrison et al., 2009*). In taste buds and many hair cell-containing epithelia, supporting cells can give rise to new sensory cells in adulthood. Similarly, touch domes are marked by a population of Keratin 17 (K17)-expressing epidermal keratinocytes that produce Merkel cells in adult skin (*Doucet et al., 2013*; *Moll et al., 1993*; *Woo et al., 2010*). An intriguing recent study indicates that Merkel cells and primary hair follicles derive from a common placode during embryogenesis (*Nguyen et al., 2018*). Interestingly, these placode-derived structures are innervated by distinct types of sensory neurons in mature skin: primary (or guard) hair follicles are innervated by rapidly adapting mechanosensory afferents whereas touch domes show selective innervation by slowly adapting afferents that express TrkC, a neurotrophin receptor encoded by the *Ntrk3* gene (*Bai et al., 2015*; *Li et al., 2011*). The developmental mechanisms through which the touch dome emerges as a structure distinct from the hair follicle and recruits appropriate sensory innervation are unknown.

We hypothesize that touch domes co-opt placode signaling mechanisms to build specialized touch receptors in discrete areas of skin. This model predicts that touch domes, like sensory placodes, contain co-clustered epithelial and mesenchymal cell types and recruit specific sensory innervation. To test these predictions, we analyzed mouse touch-dome development during embryogenesis.

## Results

### Mouse touch-dome epithelia emerge as distinct structures at E16.5

We first sought to identify epithelial cell clusters whose localization marks developing touch domes. In hair follicles, K17 expression turns on in placodes and persists in a subset of keratinocytes into adulthood (*Figure 1A*; *Bianchi et al., 2005*). By analogy, we postulated that K17 might mark nascent touch domes during embryogenesis, given that columnar keratinocytes in mature touch domes are K17 positive (*Doucet et al., 2013*; *Moll et al., 1993*). To test this hypothesis, dorsal skin specimens were labeled with antibodies against K17 and the Merkel-cell marker K8 (*Vielkind et al., 1995*) during skin development. At E15.5, most K8-positive Merkel cells associated with K17 expression in the invaginating epithelial compartment of primary hair follicles (*Figure 1B–C*, *Figure 1—figure supplement 1* and *Figure 1–video 1*). In reconstructions of full-thickness skin specimens, low levels of K17 immunoreactivity were observed next to primary hair pegs (*Figure 1C*, *Figure 1—figure supplement 1* and *Figure 1–video 1*). At E16.5, K17-positive cells were observed in primary follicles and placodes of secondary hair follicles. Additionally, primary follicles were juxtaposed to clusters of K8-positive Merkel cells interspersed with epithelial cells that stained robustly for K17. The location and arrangement of these structures recapitulated postnatal touch domes (*Figure 1B–C*).

To better understand the patterning of Merkel cells at these stages, we quantified Merkel cells within primary follicles and developing touch domes. Full-thickness, cleared skin specimens were imaged with confocal microscopy and analyzed in three dimensions. We found that the number of Merkel cells within primary hair follicles fluctuated between E15.5 and P0 (*Figure 1D*). By contrast, the number of Merkel cells in touch domes increased steadily over these stages (*Figure 1E*). As a

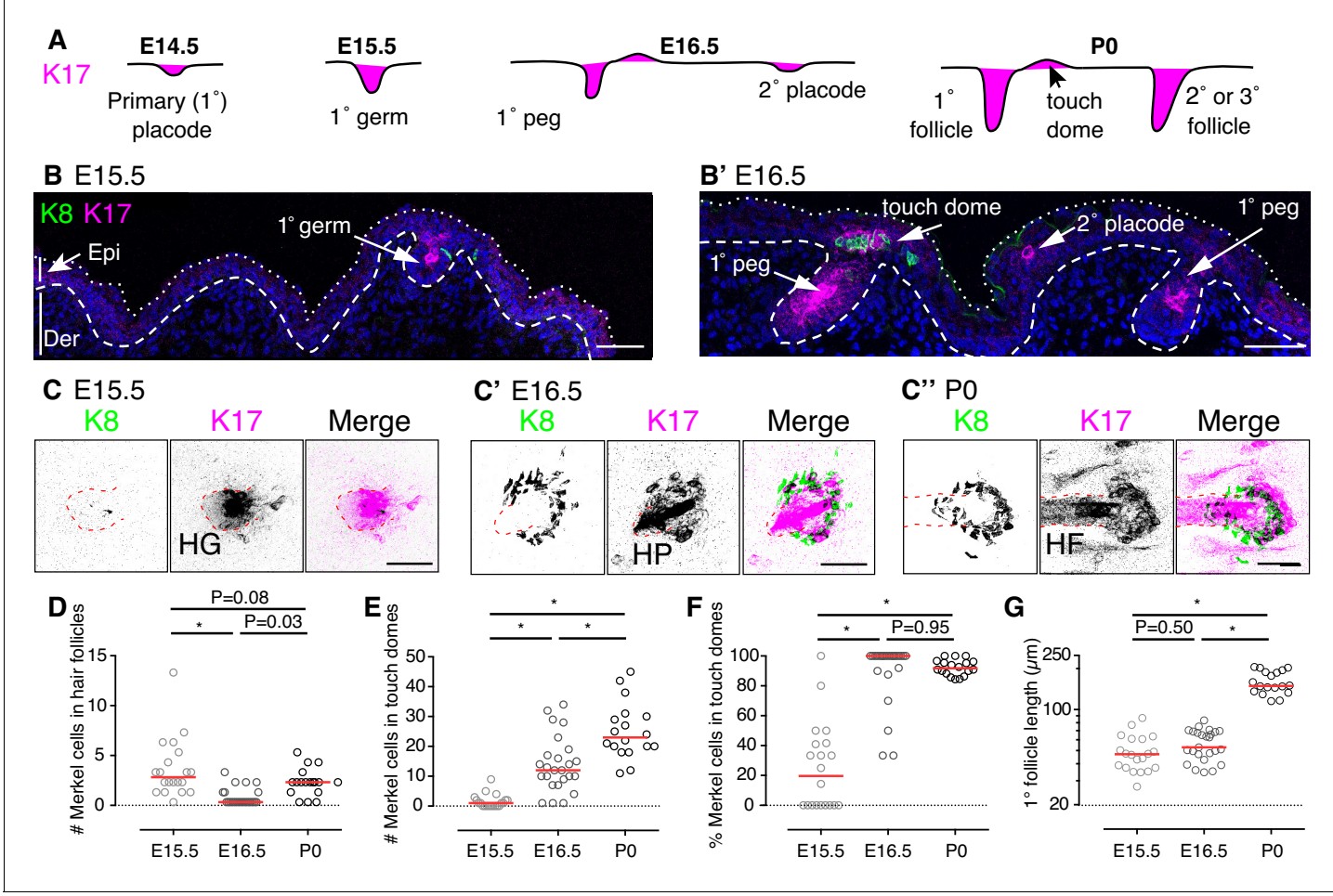

**Figure 1.** Touch domes emerge at E16.5. (**A**) Stages of hair-follicle and touch-dome morphogenesis. (**B**) Sagittal cryosections of dorsal skin at E15.5 and E16.5. Merkel cells are labeled with antibodies against K8 (green) and hair follicle and touch-dome keratinocytes are stained for K17 protein (magenta). Nuclei are labeled with DAPI (blue). Dotted and dashed lines outline the skin surface and basal epidermis, respectively. (**C**) Confocal axial projections show full-thickness cleared skin specimens at E15.5 (left trio of panels), E16.5 (middle trio), and P0 (right trio). K8 immunoreactivity: left panels and green in merged images; K17 immunoreactivity: middle panels and magenta in merged images. In the inverted lookup table (LUT) applied to merged images here and in *Figure 2,3,4,5,7* and *Figure 5—figure supplement 1*, black denotes co-localization of green and magenta pixels. Hair follicle structures (hair germ, HG, and hair peg, HP) are indicated by red dashed lines. (**D–G**) Quantification of Merkel-cell distributions and follicle lengths for primary hair follicles and touch domes at E15.5 (N = 20), E16.5 (N = 25) and P0 (N = 18). Red lines denote medians. Scatter plots show the number of Merkel cells present within each primary hair follicle (**D**) or adjacent touch domes (**E**), the corresponding percentage of Merkel cells in touch domes (**F**), and the lengths of reconstructed primary follicles (**G**). One-way ANOVA with Tukey's multiple comparisons test. *p<0.0001. Primary follicles associated with at least one Merkel cell were quantified from three mice per stage. Scale bars: 50 µm. See also *Figure 1—figure supplement 1* and *Figure 1–video 1*.

DOI: https://doi.org/10.7554/eLife.42633.002

The following video, source data, and figure supplement are available for figure 1:

**Source data 1.** Numerical values for data plotted in *Figure 1*.

DOI: https://doi.org/10.7554/eLife.42633.005

**Figure supplement 1.** Three-dimensional projections in different planes show that Merkel cells are located in both primary hair follicles and touch dome epidermis.

DOI: https://doi.org/10.7554/eLife.42633.003

**Figure 1—video 1.** Three-dimensional image stack of whole-mount immunofluorescence at E15.5.

DOI: https://doi.org/10.7554/eLife.42633.004

result, the majority of K8-positive Merkel cells shifted from a predominately hair-peg to a touch-dome localization between E15.5 and E16.5 (*Figure 1F*).

We wondered whether touch-dome emergence at E16.5 was linked to hair follicle development. As hair follicles mature, the epidermal compartment invaginates and elongates into the dermis. We

postulated that more mature primary follicles, as evidenced by their length, would have larger complements of Merkel cells in adjacent touch domes. At the population level, the lengths of primary follicles associated with Merkel cells were comparable between E15.5 and E16.5 (*Figure 1G*); however, the number of Merkel cells per touch dome increased by an order of magnitude between those stages (*Figure 1E*). These data demonstrate that the touch dome's competence to produce Merkel cells increases dramatically at E16.5 autonomous from follicle length.

Together, these data indicate that nascent touch domes, marked by the presence of K17-positive and K8-positive cells, appear just above primary follicles by E16.5. These structures are preceded at E15.5 by budding hair follicles that contain Merkel cells, which confirms a recent report (*Nguyen et al., 2018*). Although Merkel cells and primary follicles are first generated from common placode cells, nascent touch domes become highly Merkel-cell inductive between E15.5 and E16.5 in a manner that is independent of hair-follicle growth.

## *Bmp4*-expressing mesenchymal cells cluster near touch-dome epithelia at E16.5

A classic feature of a placode is its co-localization with morphogen-secreting mesenchymal cells; therefore, we next asked whether a mesenchymal population is found beneath touch domes at E16.5. As hair-follicle placodes signal to dermal papilla cells that produce BMP4 (*Botchkarev and Sharov, 2004*; *Millar, 2002*), we postulated that touch domes might also have a complement of *Bmp4*-expressing mesenchymal cells.

To assess *Bmp4* expression in developing touch domes, we analyzed the spatiotemporal pattern of Cyan Fluorescent Protein (CFP) expression in skin cryosections from $Bmp4^{CFP/+}$ reporter mice (*Jang et al., 2010*). We noted that some CFP expressing dermal cells showed intense immunoreactivity for CFP (CFP$^{high}$) whereas other cells had low levels of immunofluorescence (CFP$^{low}$). At E15.5, although CFP$^{high}$ expressing cells were scattered throughout the deep dermis, they were most prominent in dermal condensates, which are known to form early in developing hair follicles (*Figure 2A*). By E16.5 and even more obviously at E18.5, many of the brightest CFP$^{high}$ dermal cells were clustered at the hair follicle-epidermal junctures in close proximity to the Merkel-cell clusters (*Figure 2B,C*). These results were confirmed in axial projections of whole-mount skin specimens, which recapitulated live-cell imaging studies that revealed rings of CFP$^{high}$ cells around embryonic hair follicles (*Figure 2A'–C'*; *Jang et al., 2010*). Interestingly, CFP immunoreactivity was observed both in the stroma and in touch-dome keratinocytes at adult but not embryonic stages (*Figure 2D, D'*). Together, these data identify a previously unsuspected population of *Bmp4*-expressing mesenchymal cells that co-cluster with touch-dome epithelial cells.

## BMP signaling is required for touch-dome development

Given that *Bmp4*-expressing mesenchymal cells cluster concomitantly with emerging touch domes, we next sought to test the role of BMP signaling in touch-dome development. We disrupted BMP signaling by inducing epidermal expression of Noggin, a BMP inactivator, either before or after *Bmp4*-expressing mesenchymal cells first appear under touch domes. Epidermal specific expression was achieved by ultrasound-guided *in utero* delivery of lentiviruses harboring a PGK-H2BGFP marker and a doxycycline-inducible *Noggin* transgene into pregnant $Krt14^{rtTA}$ dams (*Lu et al., 2016*; *Park et al., 2006*). After viral injection into amniotic sacs at E9.5 of gestation, doxycycline chow was fed to pregnant dams starting at either E14.5 or E17.5 and continuing to P0 (*Figure 3A–B*). To test the effect of BMP disruption on touch domes, GFP-positive skin specimens (P0) were labeled with antibodies against K17, K8 and Neurofilament H (NFH), which is a marker of myelinated large-caliber afferents including the slowly adapting mechanoreceptors that innervate Merkel cells (*Figure 3C–F*).

Consistent with results from *Krt14*-driven constitutive Noggin overexpression (*Plikus et al., 2004*), primary guard hair follicles were readily apparent at P0 when epidermal Noggin expression was induced at E14.5 and E17.5. In striking contrast, however, touch-dome development was dramatically disrupted when epidermal Noggin expression was induced at E14.5. K17-positive area, Merkel-cell number and innervation area were decreased as compared with controls (*Figures 3C–D, G–I*). Interestingly, when Noggin expression was induced at E17.5, Merkel-cell numbers were similar to their control counterparts (*Figures 3E–F,J–L*).

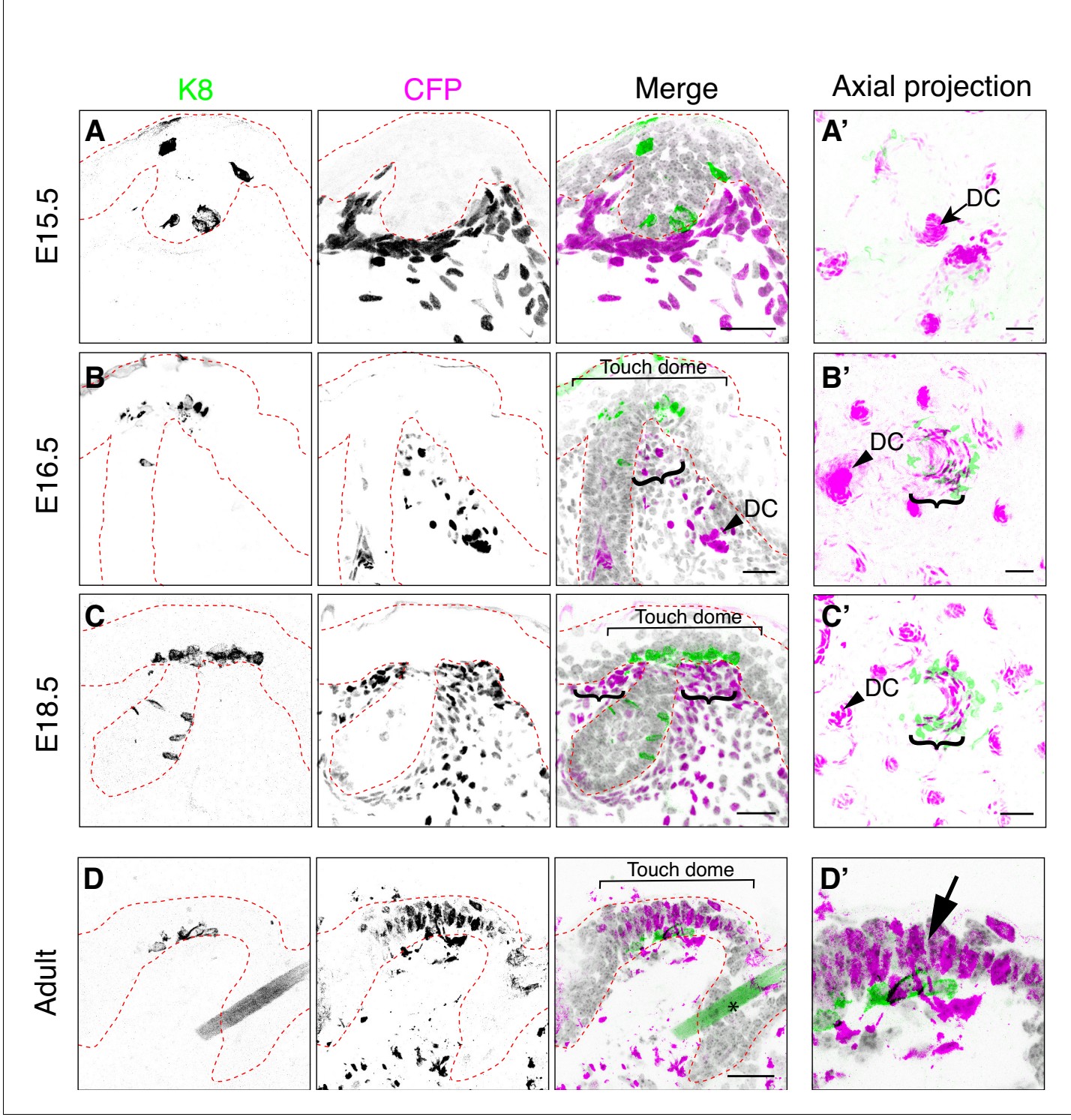

**Figure 2.** *Bmp4*-expressing cells form mesenchymal condensates under touch-dome placodes. Representative images of immunohistochemistry in *Bmp4^CFP* mice. (A–D, D') Sagittal cryosections of dorsal skin at E15.5 (A), E16.5 (B), E18.5 (C) and adult (9 month old female; D, D'). Anti-K8 antibodies labeled Merkel cells (left panels and green in merged images). Anti-GFP antibodies labeled CFP-expressing cells (middle panels and magenta in merged images). DAPI is shown in grayscale. Red dashed lines outline epidermis. Asterisk (D) indicates an autofluorescent hair shaft in adult skin. In (B–C), brackets denote touch domes, curly brackets indicate clusters of CFP^high cells beneath touch domes, and arrowheads indicate CFP^high cells in dermal condensates (DC). In (D'), arrow indicates epidermal expression of CFP driven from the *Bmp4* locus in adult epidermis. (A'–C') Confocal axial projections of full-thickness dorsal skin at E15.5 (A'), E16.5 (B') and E18.5 (C'). Pseudocolor in merged images indicates K8 (green) and K17 staining

*Figure 2 continued on next page*

*Figure 2 continued*

(magenta). In axial projections, DCs and dermal papilla (DP) appear as intensely stained dots and touch dome mesenchymal clusters appear as crescents. Images are representative of touch domes from 2–4 animals per stage. Scale bars: 25 μm.

DOI: https://doi.org/10.7554/eLife.42633.006

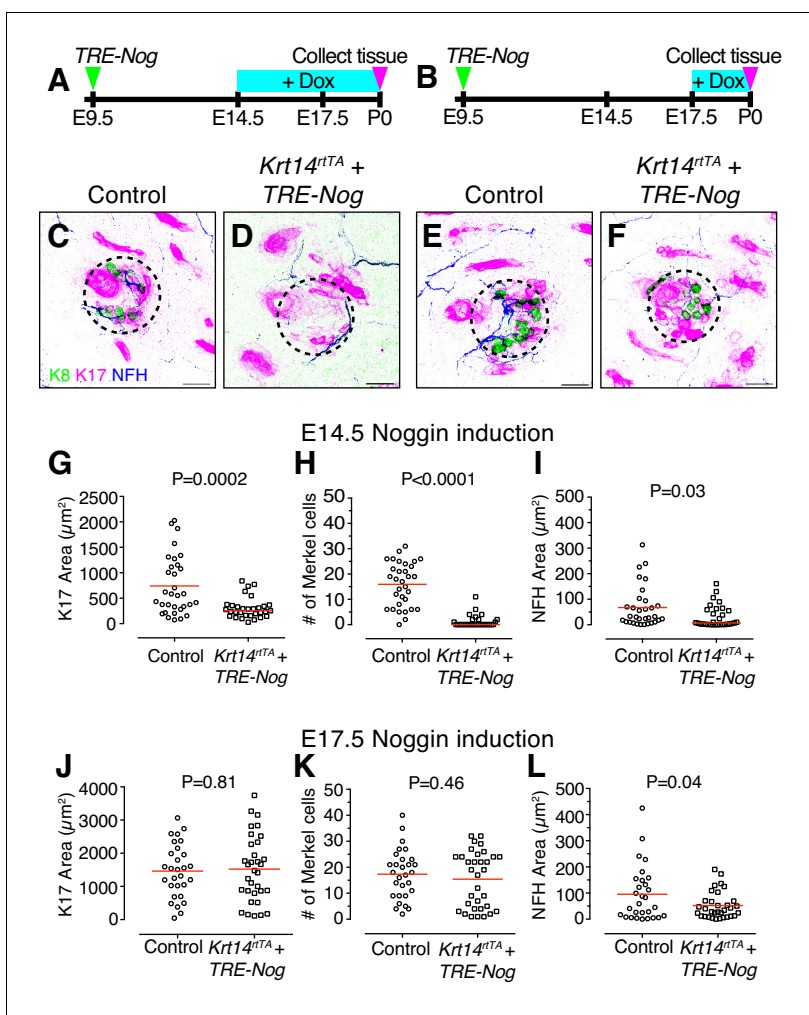

**Figure 3.** BMP signaling is required for touch-dome development. (**A–B**) Experimental design. (**C–F**) Representative axial projections of P0 dorsal skin labeled with K8 (green), K17 (magenta) and NFH (blue) after doxycycline (Dox)-induction of Noggin expression from E14.5–P0 (**C–D**) or E17.5–P0 (**E–F**). Dashed circles denote touch domes. (**C, E**) Control mice either lacked the *Krt14*-driven Tet activator transgene (*TRE-Nog* + Dox) or the virally induced Tet responsive Noggin transgene (*Krt14^{rtTA}* + Dox). (**D, F**) Experimental mice overexpressed Noggin in epidermis (*Krt14^{rtTA}* + *TRE-Nog* + Dox). (**G–I**) Quantification of K17 area (**G**) number of Merkel cells per touch dome (**H**), and area of NFH immunoreactivity (**I**) for induction from E14.5–P0 (control, N = 32 touch domes; Noggin overexpression, N = 31 touch domes). (**J–L**) The same measures from litters induced with Dox at E17.5–P0 (control, N = 29 touch domes; Noggin overexpression, N = 33 touch domes). Red lines denote means. Statistical significance was assessed with Student's *t* tests (two-tailed). Three animals per group were analyzed. Scale bars: 25 μm.

DOI: https://doi.org/10.7554/eLife.42633.007

The following source data is available for figure 3:

**Source data 1.** Numerical values for data plotted in *Figure 3*.

DOI: https://doi.org/10.7554/eLife.42633.008

The distinct effects of E14.5 and E17.5 Noggin induction on touch-dome formation could be due to differences in the duration of Noggin expression. Alternatively, these findings could reflect a specific requirement for BMP signaling in touch-dome development at E14.5–E17.5, which is the critical window during which touch domes emerge.

### NFH-positive afferents initiate Merkel-cell contacts at 16.5

To transmit coherent information to the central nervous system, sensory epithelia become selectively innervated by distinct complements of sensory neurons during development. Thus, we next sought to define the temporal dynamics of touch-dome innervation from E15.5 to P0. Dorsal skin specimens were stained for K8 and NFH. Confocal axial projections were depth-coded on a pseudocolor scale to distinguish superficial and deep structures in skin (*Figure 4A–D*). Cryosections were also used to assess the localization of afferent terminals to epidermis versus dermis (*Figure 4A'–D'*). At E15.5, NFH-positive branches protruded from the dermal plexus to upper dermal regions but did not contact Merkel cells (*Figure 4A,A'*). By E16.5, NFH-positive branches projected into the epidermis and contacted Merkel cells (*Figure 4B,B'*). From E18.5 to early postnatal stages, NFH-positive axons robustly innervated touch domes, extending branches superficially and laterally to Merkel cell clusters (*Figure 4C–D'*). Other epidermal areas lacked NFH-positive branches, indicating that these

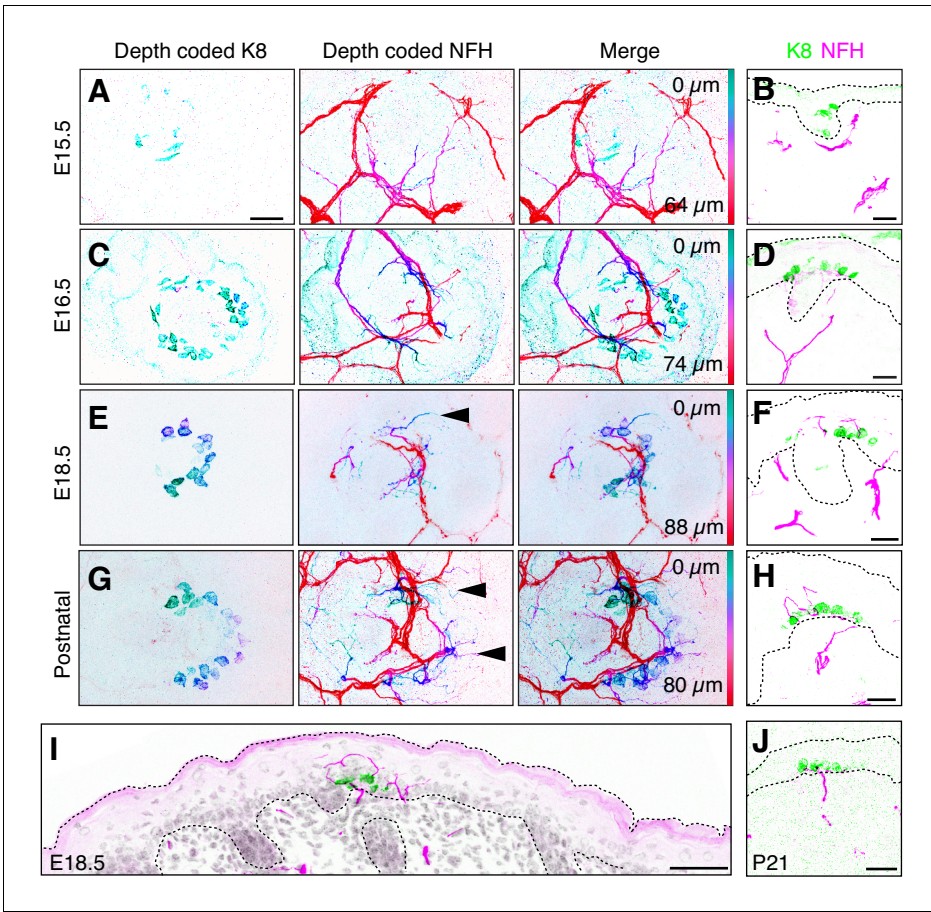

**Figure 4.** NFH-expressing afferents initiate Merkel-cell contacts at E16.5 and overshoot Merkel cells at E18.5. (A–D) Depth-coded axial projections of full-thickness dorsal skin. Teal-green structures are located near the skin surface whereas red structures are embedded deep in the dermis. For each stage, the specific Z-depth code is indicated by color scale bars at right. Black scale bar in (A) (25 µm) applies also to (B–D). (A'–D'), (E–F) Sagittal cryosections of developing skin demonstrate the time course of Merkel-cell innervation. K8: green, NFH: magenta. Scale bars: 50 µm. Dashed lines outline the epidermis. All images are representative of data from 2 to 4 animals per stage.

DOI: https://doi.org/10.7554/eLife.42633.009

neurons selectively target touch domes at this stage (*Figure 4E*). By P21, NFH-positive branches terminated on the inferior aspect of Merkel cells, which is typical of mature touch-dome afferents (*Figure 4F*). Together, these data suggest that innervation begins when touch domes emerge at E16.5 and implicate E18.5 as a key developmental time point marked by exuberant neuronal sprouting in touch domes.

## Two distinct NFH-positive afferent populations innervate embryonic touch domes

Given the unanticipated pattern of touch-dome innervation at E18.5, we next sought to determine the molecular identity of NFH-positive neurons innervating embryonic touch domes. Merkel-cell afferents are of the TrkC (*Ntrk3*) lineage; therefore, we analyzed touch-dome innervation in *Ntrk3$^{tdTomato}$* reporter mice, which mark Merkel-cell afferents, circumferential endings around hair follicles, and a population of NFH-negative afferents in mature skin (*Bai et al., 2015*). In embryonic dorsal root ganglia (DRG), 98 ± 2% of tdTomato-positive neurons were immunoreactive for NFH (mean ±SD; N = 3 mice, with 189–372 tdTomato-positive neurons quantified per mouse; *Figure 5—figure supplement 1*), which is consistent with peripheral labeling patterns observed in adult animals (*Bai et al., 2015*). Among NFH-positive DRG somata, 23 ± 3% lacked tdTomato immunoreactivity (mean ±SD; N = 3 mice, with 230–472 NFH-positive neurons quantified per mouse; *Figure 5—figure supplement 1*). Thus, this *Ntrk3$^{tdTomato}$* reporter strain identifies two molecularly distinct populations of NFH-positive neurons in embryonic DRGs.

We next analyzed the peripheral innervation patterns of NFH-positive and tdTomato-positive sensory neurons. Surprisingly, analysis of E18.5 tissue revealed that NFH-positive afferents in touch domes included both tdTomato-positive and tdTomato-negative afferents (*Figure 5A*). TdTomato-positive afferents terminated onto Merkel cells; however, tdTomato-negative terminals extended beyond Merkel-cell clusters into the touch dome's spinous and granular layers (*Figure 5A–B*). By P21, when sensory neurons are functionally mature (*Koltzenburg et al., 1997*), NFH and tdTomato immunoreactivity co-localized completely and NFH immunoreactivity was not observed superficial to the Merkel-cell layer (*Figure 5C*; *Figure 5—figure supplement 2*). Together, these results suggest that NFH-positive, tdTomato-negative afferents innervate touch domes transiently during development, whereas TrkC lineage innervation persists in adult touch domes.

The density and kinetics of tdTomato-positive and tdTomato-negative innervation in touch domes was next quantified. For line-crossing analysis, parallel lines were laid across each image to measure branch crossings spanning the touch-dome epidermis at E18.5 (*Figure 5D,E*). The frequency of tdTomato-negative branch crossings was nearly threefold that of tdTomato-positive afferents. These data indicate that tdTomato-negative branches course through the embryonic touch dome more densely than tdTomato afferents. Moreover, tdTomato-negative endings extended superficially, with 75% of branches terminating 18 µm below the epidermal surface, whereas 75% of tdTomato-positive endings terminated at the Merkel-cell layer, 27 µm below the epidermal surface (*Figure 5E*). The time course of Merkel-cell contact by both tdTomato afferents and NFH-positive branches fit well with a single exponential relation (τ = 5.1 d; *Figure 5F*). Given that these neurons differed in their molecular and morphological features, we conclude that two distinct neuronal populations innervate touch domes with similar kinetics during development.

## Touch-dome keratinocytes, but not Merkel cells, pattern sensory innervation

We next sought to define the role of epidermal cell types in establishing touch-dome innervation. *Krt17$^{CreERT2/+}$*;R26$^{DTA}$ mice, which express Diphtheria toxin (DTA) at the Rosa26 (R26) locus in K17-lineage cells after exposure to Tamoxifen, were used to ablate K17-positive touch-dome keratinocytes in embryogenesis (*Figure 6A*). Pregnant dams were injected with Tamoxifen from E15-E17, a window during which primary hair pegs have already formed but that is critical for touch-dome development (*Figure 3*). To assess touch-dome architecture, dorsal skin samples from *Krt17$^{CreERT2/+}$*;R26$^{DTA}$ ablated and littermate R26$^{DTA}$ control mice were stained in whole mount with antibodies against K17, K8 and NFH to evaluate touch-dome keratinocytes, Merkel cells and sensory afferents, respectively (*Figure 6B–C*).

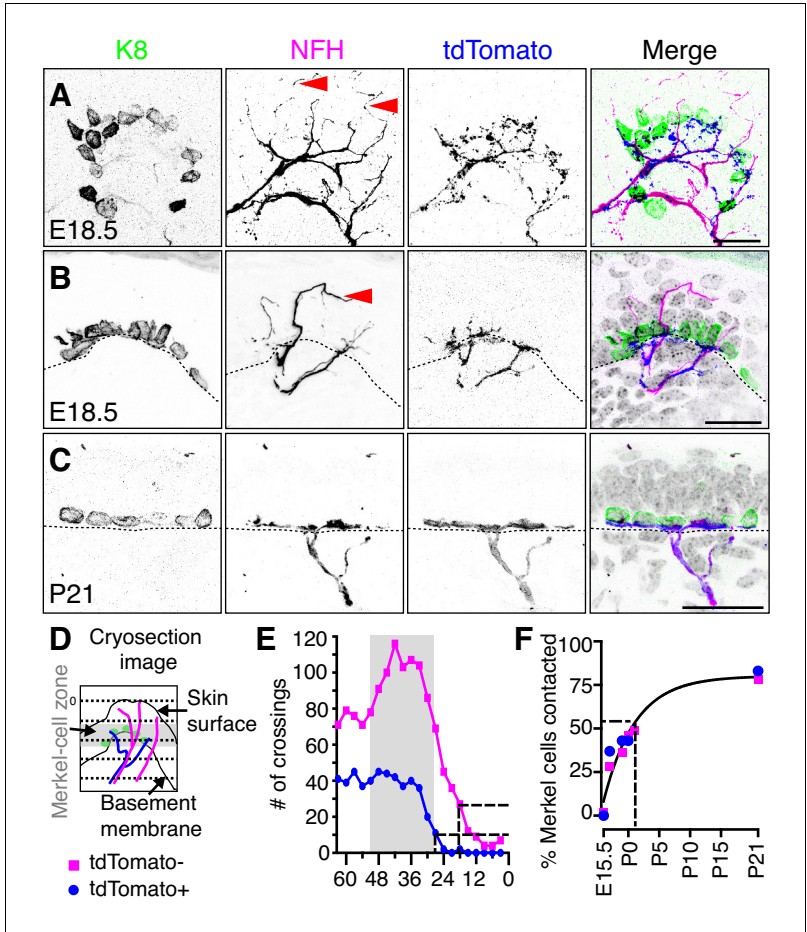

**Figure 5.** Two molecularly distinct populations of NFH-positive neurons transiently innervate touch domes during development. (**A**) Axial projections of full-thickness skin specimens from a *Ntrk3*^*tdTomato* transgenic reporter mouse at E18.5 reveal that Merkel-cell clusters (left panel and green in merge) are contacted by both tdTomato-negative afferents (arrowheads; second panel and magenta in merge) and tdTomato-positive afferents (third panel and blue in merge). (**B**) Sagittal sections also show that tdTomato-negative afferents (arrow) innervate superficial layers of touch domes at E18.5. (**C**) TdTomato and NFH immunoreactivity overlaps at P21. Color scheme in (**B–C**) is as in **A**). Dashed lines (**B, C**) denote the dermal-epidermal border. (**D**) Schematic of line crossing analysis. A series of lines was overlaid on each image starting from the skin surface and continuing toward the dermis at 3 μm intervals. The number of times each marker (tdTomato or NFH) crossed a line was counted. (**E**) Quantification of line crossings. Gray box indicates the span of the Merkel-cell bearing epidermal zone. Dashed lines indicate the distance from the surface at which the line crossing curve decays by 75% (N = 50 touch domes from three animals). (**F**) The proportion of Merkel cells contacted by tdTomato-positive afferent branches (N = 35–387 Merkel cells from 2 to 3 mice per stage) and tdTomato-negative afferent branches (N = 87–521 Merkel cells from 2 to 6 mice per stage) were plotted versus developmental stage. Both datasets were well fit by a single exponential (black line; τ = 5.1 d; $R^2$ = 0.9), which indicates that most Merkel cells are innervated by P1. See also *Figure 5—figure supplement 1*.

DOI: https://doi.org/10.7554/eLife.42633.010

The following source data and figure supplements are available for figure 5:

**Source data 1.** Numerical values for data plotted in *Figure 5*.
DOI: https://doi.org/10.7554/eLife.42633.014

**Figure supplement 1.** Few tdTomato-positive DRG cell bodies lack NFH expression in E16.5*Ntrk3*^*tdTomato*mice.
DOI: https://doi.org/10.7554/eLife.42633.011

**Figure supplement 1—source data 1.** Numerical values for data plotted in *Figure 5—figure supplement 1*.
DOI: https://doi.org/10.7554/eLife.42633.012

**Figure supplement 2.** NFH-expressing afferent endings terminate at the basal epidermis at P21.
DOI: https://doi.org/10.7554/eLife.42633.013

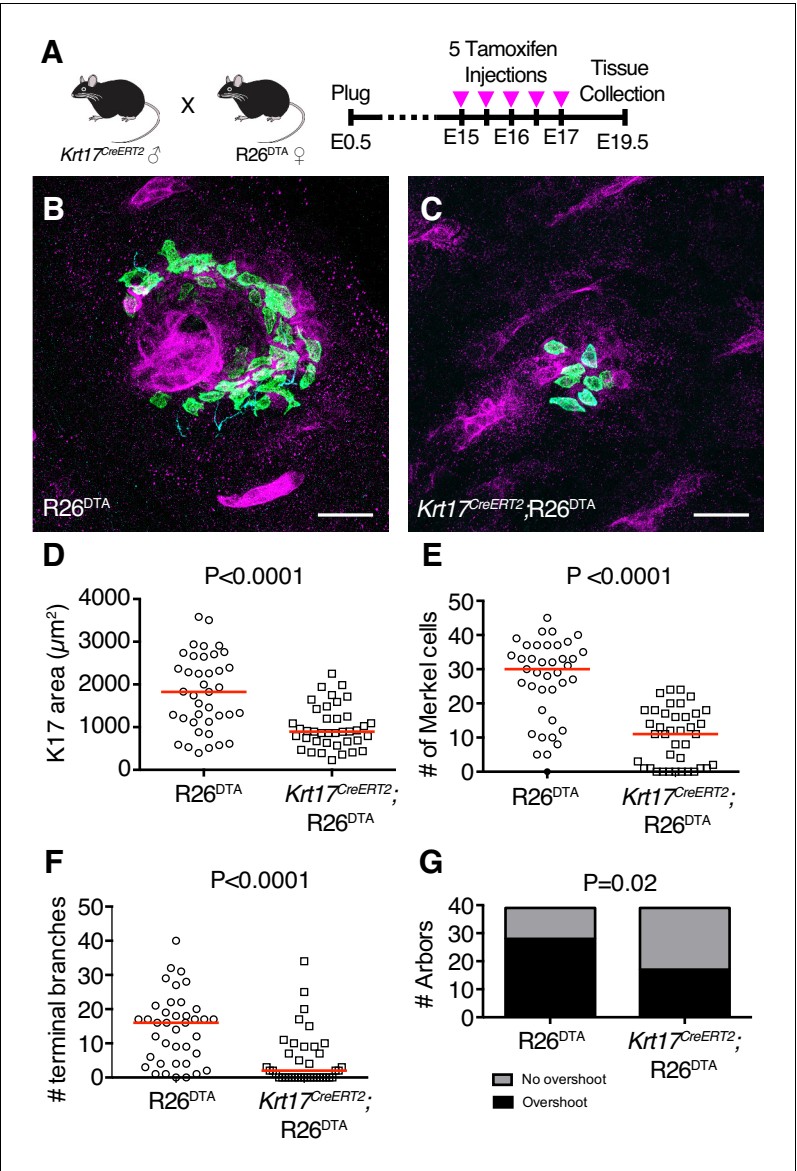

**Figure 6.** K17-lineage keratinocytes are necessary for patterning touch-dome afferents during embryogenesis. (A) Experimental design. (B–C) Axial projections of full-thickness E19.5 skin specimens from R26$^{DTA}$ and Krt17$^{CreERT2}$; R26$^{DTA}$ mice labeled with antibodies against K8 (green), K17 (magenta) and NFH (cyan). (D–G) Quantification of K17 area (D), number of Merkel cells (E), number of terminal branches (F), number of arbors exhibiting sprouting above Merkel cells (G), across genotypes for mice that were administered Tamoxifen from E15–E17. N = 39 touch domes from three animals per group. Red lines denote medians. Scale bars: 25 μm. Statistical significance was assessed with a Student's t-test in (D), Mann-Whitney tests in (E–F) and a Fisher's exact test in (G). See also *Figure 6—figure supplements 1* and *2*.

DOI: https://doi.org/10.7554/eLife.42633.015

The following source data and figure supplements are available for figure 6:

**Source data 1.** Numerical values for data plotted in *Figure 6*.
DOI: https://doi.org/10.7554/eLife.42633.018

**Figure supplement 1.** K17-lineage keratinocytes are successfully ablated in embryonic Krt17$^{CreERT2}$;R26$^{DTA}$ mice.
DOI: https://doi.org/10.7554/eLife.42633.016

**Figure supplement 2.** K17-lineage keratinocytes are sufficient to recruit NFH-expressing afferents to the epidermis.
DOI: https://doi.org/10.7554/eLife.42633.017

K17 area was markedly reduced in hair follicles and touch domes from $Krt17^{CreERT2}$;R26$^{DTA}$ mice as compared to R26$^{DTA}$ mice, indicating that ablation was successfully achieved (*Figure 6D* and *Figure 6—figure supplement 1*). Merkel-cell numbers were also reduced in K17-lineage ablated mice (*Figure 6E*). The median number of NFH-expressing terminal endings decreased from 16 in control mice to two in ablated mice, demonstrating that touch-dome epidermal cells are required to establish innervation at late embryonic stages (*Figure 6F*). The number of touch-dome afferents that extended branches superficial to Merkel cells was also decreased in K17-lineage ablated mice (*Figure 6G*). Given the dramatic loss of sensory branches and the reduction in the proportion of arbors that overshoot the Merkel-cell layer, we conclude that touch-dome epidermal cells are necessary to establish the terminal architecture of NFH-positive afferents.

To test whether K17-expressing keratinocytes are sufficient to recruit NFH-positive innervation, we developed a strategy to induce proliferation of this cell type. Overexpression of epidermal Sonic Hedgehog (Shh) causes hyperplastic expansion of K17-positive keratinocytes in adult mice without altering Merkel-cell numbers (*Xiao et al., 2015*). Thus, we used an *in utero* lentiviral strategy to ectopically express *Shh* in the embryonic epidermis (*Figure 6—figure supplement 2*). *Shh* expression was induced from E16.5 to E18.5 to avoid disruption of primary hair-follicle development (*Lu et al., 2016*; *Perdigoto et al., 2016*). Consistent with the effects of Shh overexpression in adult epidermis, K17-expressing keratinocytes were markedly expanded in GFP-positive skin areas at early postnatal stages. This expansion of the K17-positive epidermal compartment was accompanied by an increase in NFH-positive skin innervation (*Figure 6—figure supplement 2*). Although Shh induction at E12.5 induces supernumerary Merkel cells (*Nguyen et al., 2018*; *Perdigoto et al., 2016*), ectopic Merkel cells were not observed in our experimental paradigm. Together, these results suggest that K17-expressing keratinocytes rather than Merkel cells recruit NFH-positive innervation.

To directly test whether Merkel cells are required for embryonic touch-dome innervation, we reconstructed and traced tdTomato-positive and tdTomato-negative afferents in *Atoh1* null mice, which lack Merkel cells (*Ben-Arie et al., 2000*; *Maricich et al., 2009*). As *Atoh1* null mutants die at birth, dorsal skin specimens from *Atoh1* null mice ($Ntrk3^{tdTomato/+}$;$Atoh1^{LacZ/LacZ}$) and littermate controls ($Ntrk3^{tdTomato/+}$;$Atoh1^{LacZ/+}$) were assessed at E18.5 by labeling with antibodies against tdTomato, NFH and K8 (*Figure 7A–B*). The structure of touch-dome afferents, as measured by the number of terminal branches, highest branch order, and complexity index, did not differ between genotypes (*Figure 7C–E*). Thus, Merkel cells are not required for proper touch-dome innervation during development.

Together, these results identify K17-positive keratinocytes as a critical epithelial cell type that is both necessary and sufficient for patterning touch-dome afferents during embryogenesis.

## Discussion

Building mammalian touch receptors presents the challenge of pairing dynamic epithelial structures with their appropriate complement of sensory afferents. In the case of Merkel cell-neurite complexes, tremendous strides have been made to identify both molecular mechanisms that specify Merkel cells and signaling cascades that give rise to distinct populations of somatosensory neurons (*Morrison et al., 2009*; *Nguyen et al., 2018*; *Perdigoto et al., 2014*; *Xiao et al., 2016*). Here, we propose that touch domes co-opt evolutionarily conserved placodal mechanisms to assemble these mechanosensory complexes.

Placodes are patches of epithelial cells that give rise to sensory structures and epidermal appendages such as hair follicles, glands and feathers (*Biggs and Mikkola, 2014*; *Piotrowski and Baker, 2014*; *Schlosser, 2005*; *Streit, 2004*). In concert with co-clustered mesenchymal cell types, placodes generate sensory epithelia that are selectively innervated. Here, we demonstrate that Merkel cells and K17-positive keratinocytes are spatiotemporally coordinated with a previously undescribed condensate of *Bmp4*-expressing mesenchymal cells in embryonic skin (*Figure 8A*). Moreover, our studies define a window during embryogenesis during which BMP signaling is required for touch-dome development (*Figure 8B*). Finally, K17-lineage keratinocytes drive the selective touch-dome targeting of two molecularly and morphologically distinct populations of myelinated afferents (*Figure 8B*). Collectively, these results identify a new role and mesenchymal source for BMPs in skin development and define cellular mechanisms that pattern sensory innervation of touch domes.

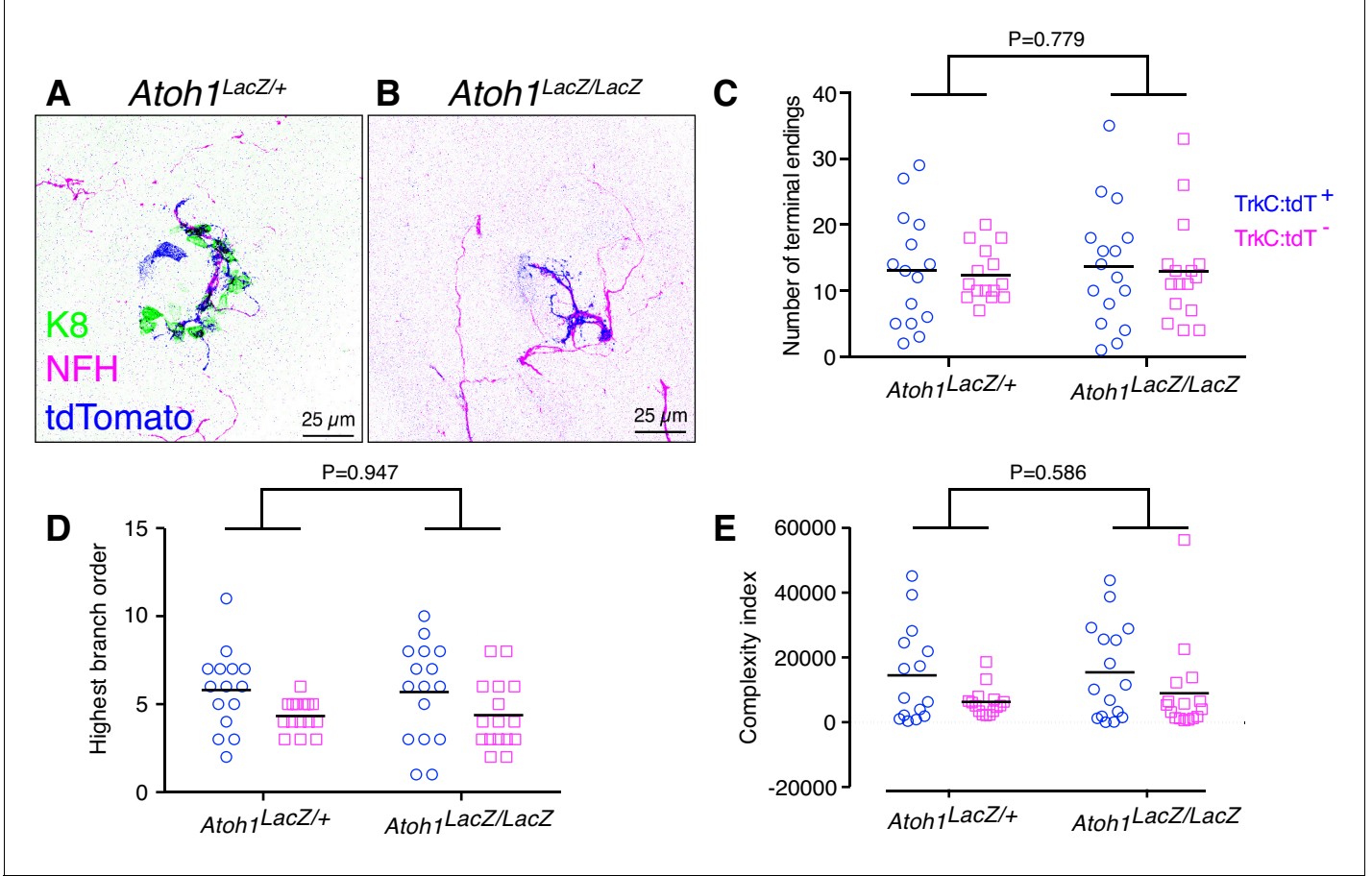

**Figure 7.** Merkel-cells are not required for embryonic touch-dome innervation. (A–B) Axial projections of whole-mount immunohistochemistry from control (A) and *Atoh1* knockout mice (B). Tissues were labeled with antibodies against K8 (Merkel cells, green), NFH (myelinated neurons, magenta) and tdTomato (Ntrk3-lineage neurons; blue). (C–E) Quantification of terminal branches per touch dome (C), highest branch order (D) and complexity index (E) for tdTomato-expressing and tdTomato-negative afferents (N = 15 control and 16 *Atoh1* knockout touch domes from four animals per group). Black lines indicate medians. By two-way ANOVA, the genotype effect was not significantly different for any measure. For the afferent type effect, tdTomato-positive and tdTomato-negative afferents showed significant differences in highest branch order (D, p=0.012) and complexity index (E, p=0.028) but number of terminal endings did not differ (C, p=0.705).

DOI: https://doi.org/10.7554/eLife.42633.019

The following source data is available for figure 7:

**Source data 1.** Numerical values for data plotted in *Figure 7*.
DOI: https://doi.org/10.7554/eLife.42633.020

## Placodes generate diverse epithelial structures

In vertebrates, numerous epidermal appendages and sensory tissues are placode-derived. Placodes give rise to epithelial sensory cells, such as taste buds and hair cells, as well as *bona fide* sensory neurons, such as olfactory neurons (*Hall et al., 1999*; *Maier et al., 2014*; *Whitfield, 2015*). The otic placode is the source of all inner-ear cell types including hair cells. Similarly, hair-cell containing neuromasts in the lateral lines of aquatic organisms develop from migrating placodes that decorate the body axis with precursor cells (*Agarwala et al., 2015*; *Dambly-Chaudière et al., 2003*; *Metcalfe et al., 1985*). A conserved feature of placodes is that epithelial and mesenchymal cells co-cluster to engage in molecular crosstalk required for tissue morphogenesis. Similarly, we observed molecularly distinct populations of epithelial and mesenchymal cells that mark nascent touch domes during embryogenesis.

Touch domes, marked by patches of K17-expressing epithelial cells interspersed with Merkel cells, emerge as anatomical structures that can be distinguished from their associated primary hair

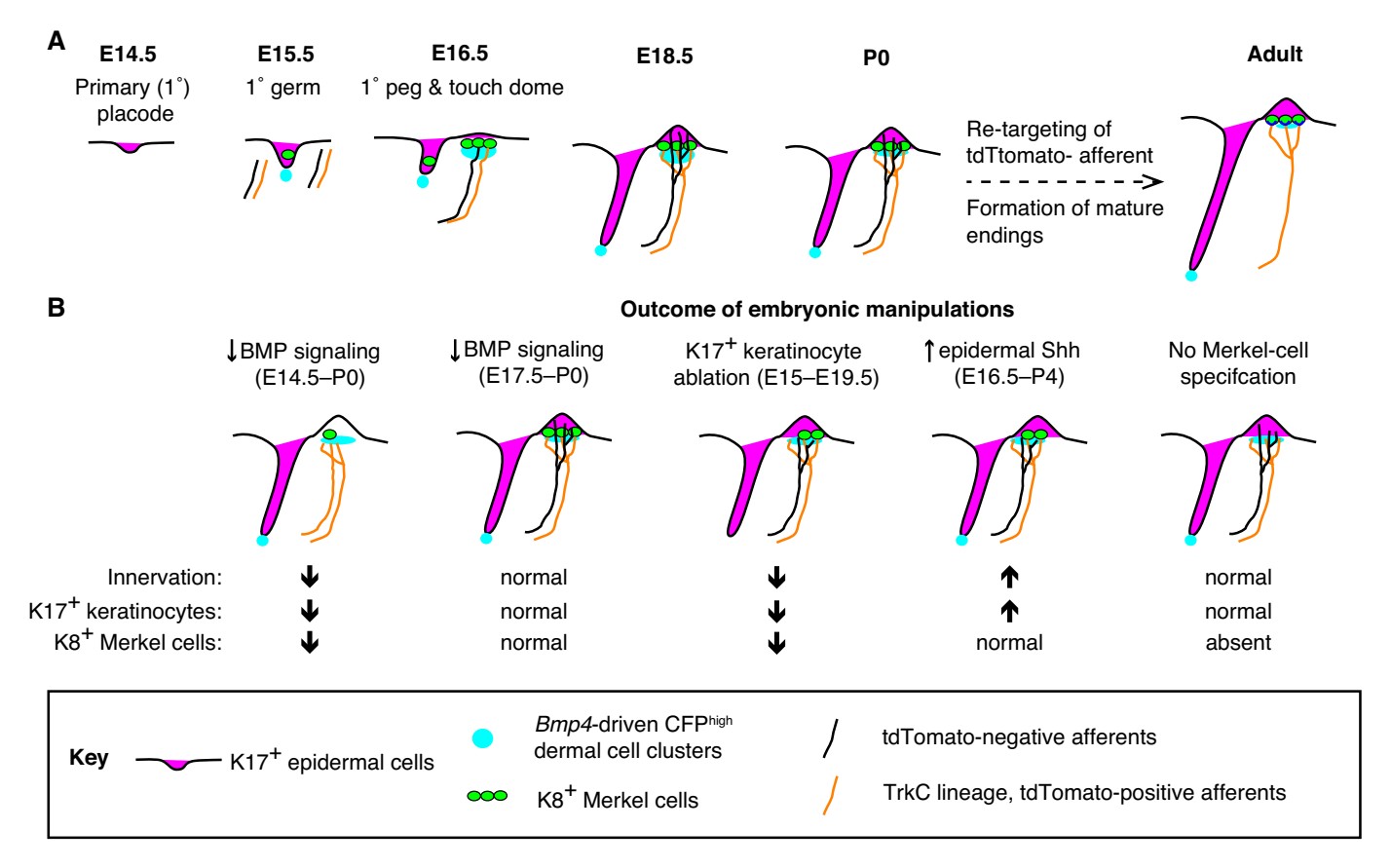

**Figure 8.** A model of touch-dome innervation during development. (**A**) The progression of developing touch domes, including the localization of epidermal K17-positive cells (magenta) and Merkel cells (green), CFP[high] dermal cells (cyan) and NFH-positive sensory afferents (black, non-TrkC lineage and orange, TrkC lineage). (**B**) Summary of experimental manipulations of touch-dome cell types and BMP signaling.
DOI: https://doi.org/10.7554/eLife.42633.021

follicles by E16.5. Consistent with a recent report, we observed that K8-positive Merkel cells localize to hair pegs at E15.5 (*Nguyen et al., 2018*). It is not surprising that hair follicles can contain Merkel cells, given that Merkel cells are observed in adult hair follicles in whisker pads, some guard hairs and distal limbs (*Woodbury and Koerber, 2007*). Moreover, knockout of *Ezh1* combined with *Krt14*[Cre] mediated *Ezh2* mutation in *polycomb* results in abundant ectopic Merkel cells in hair follicles (*Perdigoto et al., 2014*). Interestingly, Sox9-positive, Lhx2-negative primary placode cells appear to serve as progenitors of both hair follicles and touch-dome Merkel cells (*Nguyen et al., 2018*).

How do these placode cells give rise to Merkel cells that are differentially localized to hair pegs at E15.5 and touch domes at E16.5? One possibility is that K17-positive keratinocytes and Merkel cells migrate laterally from the hair peg to form touch domes. Alternatively, a common placode precursor might first produce hair follicles and then subsequently develop competence to generate touch domes. This is an attractive model for development of rodent touch domes because they are always found adjacent to primary (guard) hairs, and *tabby* mice that lack primary follicles also lack touch domes (*Vielkind and Hardy, 1996*). By contrast, touch domes are independent of hair follicles in cats and humans (*Iggo and Muir, 1969*; *Moll et al., 1993*; *Reinisch and Tschachler, 2005*). Since both touch-domes and follicle placodes derive from *Krt14*-expressing epidermal progenitors, these findings suggest that there might be species-specific differences in when touch-dome and follicle placode lineages diverge in skin development.

Simultaneous with the emergence of touch domes, *Bmp4*-expressing mesenchymal cells form underlying clusters. To our knowledge, a molecularly identified mesenchymal population has not

been previously described in the touch dome. Interestingly, in the $Bmp4^{CFP}$ reporter strain, CFP expression also marks dermal papillae, which are mesenchymal clusters required for hair-follicle morphogenesis (*Botchkarev and Sharov, 2004*). As signals from the dermal papilla trigger progression of hair-follicle development, we speculate that *Bmp4*-expressing mesenchymal cells might likewise play a role in touch-dome maturation. Consistent with this notion, we observed that CFP$^{high}$ cells tightly co-localize with touch domes during embryogenesis but then form loose networks at postnatal stages. Selective molecular markers that distinguish dermal papillae and touch-dome condensates will be needed to more fully define the role of mesenchymal signaling in touch-dome development.

## Complex signaling cascades give rise to developing Merkel cells

Development and maintenance of placode-derived structures require precise spatiotemporal control of complex signaling networks. Placodes often employ a common cast of molecular pathways, including BMPs, Wnts, Fibroblast Growth Factor, Notch and Shh (*Biggs and Mikkola, 2014*; *Streit, 2004*). For example, BMP4 is required for the development and maintenance of sensory placodes in the visual, auditory and gustatory systems (*Mistretta and Liu, 2006*; *Pujades et al., 2006*; *Sjödal et al., 2007*). In the case of Merkel cells, Eda/Edar, Wnt, Fibroblast Growth Factor, Notch and Sonic hedgehog (Shh) and Polycomb repressive complex signaling occurs upstream of Atoh1 in cell-fate specification (*Bardot et al., 2013*; *Cohen et al., 2018*; *Logan et al., 2018*; *Perdigoto et al., 2016*; *Vielkind and Hardy, 1996*; *Xiao et al., 2016*).

We focused on BMP signaling because $Bmp4^{CFP}$ reporter mice revealed *Bmp4*-expressing cells beneath developing touch domes. Postnatal mice that constitutively overexpress Noggin in epidermis have normal densities of guard hairs, which indicates that BMP signaling is not required for primary placode specification and follicle differentiation (*Plikus et al., 2004*). Our data demonstrate that BMP signaling from E14.5–P0 is required for the development of touch-dome cell types, including K17-positive keratinocytes, Merkel cells and sensory afferents. By contrast, touch dome K17-positive keratinocytes and Merkel cells were not affected when BMP signaling was disrupted from E17.5–P0. The differences in touch-dome morphology between epidermal Noggin induction at E14.5 and E17.5 suggest that E14.5–E17.5 is a critical window for BMP signaling in touch-dome development. Alternatively, it is possible that the duration rather than the timing of embryonic BMP signaling is important for touch-dome formation. Distinguishing between these models will require development of mice that allow precise temporal control of the onset and offset of BMP signaling. Moreover, given that Noggin has high affinity for multiple BMPs, Noggin overexpression is likely to have myriad effects on morphogen signaling in skin; therefore, future studies are needed to test a direct requirement for CFP$^{high}$ dermal cells in touch-dome development.

Whereas we observed a disruption in touch-dome Merkel cell-neurite complexes with Noggin overexpression, these complexes are enhanced in whisker follicles by constitutive *Krt14*-driven overexpression of Noggin and reduced by constitutive overexpression of BMP4 (*Guha et al., 2004*). This dichotomy indicates that BMPs might play divergent roles in Merkel-cell specification in touch domes and whisker follicles. Alternatively, given that *Krt14* expression begins at E9, BMP signaling might have stage-specific effects on touch-receptor development.

In mouse epidermis, a delicate spatiotemporal balance of BMP and Shh signaling determines the fate of placode-derived appendages. In particular, dorsal placodes faced with low BMP and high Shh levels give rise to hair follicles, whereas ventral placodes, which experience high BMPs and low Shh levels, produce sweat glands (*Lu et al., 2016*). Interestingly, Shh from primary follicles is required for Merkel-cell development in dorsal epidermis (*Xiao et al., 2016*). We propose that the transient localization of *Bmp4*-expressing cells at interfollicular zones creates the potential for a third embryonic niche with high BMP and high Shh levels that promotes touch-dome innervation. In this model, temporally and spatially restricted BMP4, coupled with Shh and perhaps other signals, define touch domes as sensory appendages distinct from hair follicles.

## A transiently innervating population of NFH-positive neurons

We found that at least two NFH-positive neuronal populations selectively target the touch-dome during development but only TrkC lineage neurons persist in adult touch domes. Merkel-cell afferents are the principal afferent subtype that expresses *Ntrk3* at E12.5 (*Bai et al., 2015*), which

suggests that the tdTomato-positive terminals we observe embryonically are *bona fide* slowly adapting mechanoreceptors. The functional identity of NFH-positive afferents that lack expression of the *Ntrk3:tdTomato* reporter remains unknown. These neurons might represent TrkA-dependent Ret-expressing neurons that innervate touch domes during development; however, such TrkA-dependent neurons are NFH-negative (*Niu et al., 2014*). An intriguing area for future study is to define the molecular identity of these transient touch-dome neurons, as well as the guidance cues that mediate selective targeting and retraction of touch-dome afferents during development.

Merkel cells have long been postulated to play a role in selective recruitment or refinement of touch-dome afferents; however, NFH-positive afferents target adult touch domes in the absence of Merkel cells (*Maksimovic et al., 2014*; *Maricich et al., 2009*; *Reed-Geaghan et al., 2016*). Indeed, Merkel cells are not required to recruit either population of NFH-positive afferents that we observed in this study. Touch-dome afferents branch extensively in adult Merkel cell-knockout animals, suggesting that Merkel cells aid in refinement of afferent terminals (*Maksimovic et al., 2014*; *Maricich et al., 2009*). In this study, we did not observe a difference in terminal branch number in touch domes between control and knockout animals at E18.5. This suggests that Merkel cells do not influence the terminal structure of NFH-positive neurons during embryonic development but that they are involved in refining sensory arbors postnatally.

### K17-positive keratinocytes pattern touch-dome innervation

Our findings establish K17-lineage keratinocytes as key cellular effectors of embryonic touch-dome innervation. During the touch dome's critical developmental window (E15–E17), selectively ablating K17-positive keratinocytes disrupts touch-dome structure and innervation. Conversely, ectopically inducing K17-positive keratinocytes during late embryogenesis increases NFH-positive afferents but not Merkel cells. In wildtype skin, touch-dome afferents robustly project past Merkel cells into the K17-positive keratinocyte zone at E18.5–P1, which implies that K17-positive cells drive axonal recruitment during development. Alternatively, K17-positive cells might send signals that increase branching of NFH-positive afferents in the vicinity. Interestingly, a previous report demonstrated that K17-positive keratinocytes are required to maintain touch-dome innervation in adult mice (*Doucet et al., 2013*). Thus, K17-lineage cell types are necessary and sufficient for both development and maintenance of NFH-positive afferents in touch domes.

These findings, along with recent studies of hair-follicle innervation, suggest that epidermal-directed patterning is a fundamental principle of somatosensory innervation. For example, Aδ low-threshold mechanoreceptors known as D-hair afferents are positioned during development by BDNF-expressing keratinocytes located on the caudal side of hair follicles (*Rutlin et al., 2014*). When BDNF is disrupted in the epithelial compartment, D-hair afferents lose their characteristic rostral orientation selectivity (*Rutlin et al., 2014*). Moreover, epidermal stem cells located in hair follicles produce an extracellular matrix protein, EGFL6, that influences the terminal architecture of mechanosensory lanceolate endings (*Cheng et al., 2018*). An important area for future investigation is to define the molecular cues derived from K17-lineage keratinocytes that recruit, refine and maintain touch-dome afferents in developing and mature epidermis.

## Materials and methods

**Key resources table**

| Reagent type (species) or resource | Designation | Source or reference | Identifiers | Additional information |
| --- | --- | --- | --- | --- |
| Gene (*M. musculus*) | *Bmp4$^{CFP}$* | PMID: 20336610 | MGI: 4460795 | MGI symbol: Bmp4-tm5.1Bhr; R. Berhinger laboratory, MD Anderson |
| Gene (*M. musculus*) | *Ntrk3$^{tdTomato}$* | PMID: 26687362 | MGI: 5707201; Jax: 030292 | MGI symbol: Ntrk3-tm2.1Ddg; D. Ginty laboratory, Harvard Univ. |

*Continued on next page*

*Continued*

| Reagent type (species) or resource | Designation | Source or reference | Identifiers | Additional information |
|---|---|---|---|---|
| Gene (*M. musculus*) | *Atoh1^LacZ* | PMID: 10662643 | MGI: 2155983; Jax: 005970 | MGI symbol: Atoh1-tm2Hzo; H. Zoghbi laboratory; Baylor College of Medicine |
| Gene (*M. musculus*) | *Krt14^rtTA* | PMID: 17018284 | MGI: 3762564; Jax: 008099 | MGI symbol: Tg(KRT14-rtTA)F42Efu; Fuchs laboratory |
| Gene (*M. musculus*) | *Krt17^CreERT2* | PMID: 23727240 | MGI:5523315 | MGI symbol: Tg(Krt17-cre/ERT2)1F3Dmo; D. Owens; laboratory, Columbia Univ. |
| Gene (*M. musculus*) | *Krt14^Cre* | Jackson Labs; PMID: 11044393 | MGI: 2445832; Jax: 018964 | MGI symbol: Tg(KRT14-cre)1Amc |
| Gene (*M. musculus*) | R26^DTA | Jackson Labs; PMID: 16407399 | MGI:4440748; Jax: 010527 | MGI symbol: B6; 129-Gt(ROSA) 26Sor^tm1(DTA)Mrc/J |
| Antibody | anti-Keratin-8 (K8; rat monoclonal) | Devel. Studies Hybridoma Bank | Troma-I; RRID: AB_531826 | Deposited to DSHB by Brulet, P. and Kemler, R. |
| Antibody | anti-Neurofilament-Heavy (NFH; chicken polyclonal) | Abcam | Cat# 4680; RRID: AB_304560 | (1:3000 in cryosections; 1:500 in whole mount preps) |
| Antibody | anti-dsRed (tdTomato; rabbit polyclonal) | Clontech | Clontech: 632496; RRID:AB_10013483 | (1:1000 in cryosections; 1:500 in whole mount preps) |
| Antibody | anti-Keratin-17 (K17; rabbit monoclonal) | Abcam | Cat# 109725; RRID: AB_10889888 | (1:200 in cryosections; 1:200 in whole mount preps) |
| Antibody | anti-Green Fluorescent Protein (GFP; chicken polyclonal) | Abcam | Cat# 13970; RRID: AB_300798 | (1:1000 in cryosections; 1:500 in whole mount preps) |
| Antibody | Goat polyclonal anti-rat IgG (H + L) Alexa-594 | ThermoFisher | Cat# A-11007; RRID: AB_10561522 | (1:1000 in cryosections; 1:500 in whole mount preps) |
| Antibody | Goat polyclonal anti-rat IgG (H + L) Alexa-488 | ThermoFisher | Cat# A-11006; RRID: AB_2534074 | (1:1000 in cryosections; 1:500 in whole mount preps) |
| Antibody | Goat polyclonal anti-chicken IgY (H + L) Alexa-647 | ThermoFisher | Cat# A-21449; RRID: AB_2535866 | (1:1000 in cryosections; 1:500 in whole mount preps) |
| Antibody | Goat polyclonal anti-chicken IgY (H + L) Alexa-488 | ThermoFisher | Cat# A-11039; RRID: AB_2534096 | (1:1000 in cryosections; 1:500 in whole mount preps) |
| Antibody | Goat polyclonal anti-rabbit IgG (H + L) Alexa-594 | ThermoFisher | Cat# A-11037; RRID: AB_2534095 | (1:1000 in cryosections; 1:500 in whole mount preps) |
| Antibody | Goat oligoclonal anti-rabbit IgG (H + L) Alexa-647 | ThermoFisher | Cat# A-27040; RRID: AB_2536101 | (1:1000 in cryosections; 1:500 in whole mount preps) |
| Genetic reagent | *pLKO-TRE-Nog-PGK-H2BGFP* | PMID: 28008008 | | Fuchs laboratory |

*Continued*

| Reagent type (species) or resource | Designation | Source or reference | Identifiers | Additional information |
|---|---|---|---|---|
| Genetic reagent | *pLKO-TRE-Shh-PGK-H2BGFP* | PMID: 24813615 | | Fuchs laboratory |

## Mice

*Bmp4$^{CFP}$*, *Ntrk3$^{tdTomato}$*, *Atoh1$^{LacZ}$*, *Krt14$^{rtTA}$*, *Krt14$^{Cre}$*, *Krt17$^{CreERT2}$* and R26$^{DTA}$ animals were described previously (*Bai et al., 2015*; *Ben-Arie et al., 2000*; *Dassule et al., 2000*; *Doucet et al., 2013*; *Jang et al., 2010*; *Nguyen et al., 2006*; *Wu et al., 2006*). Animal studies were approved by the Institutional Animal Care and Use Committees of Columbia University and The Rockefeller University. Specimens were collected from both embryonic and postnatal mice. Gestation day E0.5 was defined as 12PM on the day a plug was discovered. Both male and female mice were used for all experiments.

## Immunohistochemistry

Dorsal skin samples or whole embryo bodies were fixed with 4% paraformaldehyde on ice for three hours. For whole–mount preparations, skin samples were washed in Triton-X PBS (PBS-T) for several hours and then incubated in primary and secondary antibodies. Whole-mount immunohistochemistry was performed as previously described (*Lesniak et al., 2014*; *Li et al., 2011*). Briefly, tissue was fixed in 4% paraformaldehyde (PFA) overnight, washed in 0.01% (for samples < E18.5) or 0.03% (for samples ≥ E18.5) PBS-T and incubated in primary antibody for 72–96 hr at 4°C. After 5–10 hr of washes in PBS-T, samples were incubated for 48 hr at 4°C in secondary antibodies. After staining, tissue was dehydrated in a series of methanol dilutions (25–100%) and cleared using a 2:1 benzyl benzoate/benzyl alcohol solution.

For histochemistry cryosections, samples were cryoprotected after fixation for 48 hr at 4°C in 30% (wt/vol) sucrose. Immunohistochemistry was performed as previously described (*Lumpkin et al., 2003*). Briefly, tissue was first incubated in 0.01% PBST 5% NGS for 1 hr at room temperature and then in primary antibodies overnight at 4°C. After washing in PBS, samples were incubated for 45 min at room temperature in secondary antibodies. Tissue was then washed five times in PBS and mounted with Fluoromount.

## Imaging and image processing

All specimens were imaged in three dimensions (0.5–1 µm axial step sizes) on a Zeiss Exciter confocal microscope equipped with 20X, 0.8 NA and 40X, 1.3 NA objective lenses or on a Nikon A1 confocal microscope outfitted with 20X, 0.75NA and 40X, 1.3NA lenses. Quantification was performed in unprocessed images. For publication, representative images were cropped to regions of interest and output levels were linearly adjusted to ensure the histogram filled the dynamic range. Gamma was not changed. In *Figures 1*, *2*, *3*, *5* and *7*, images are displayed using an inverted lookup table (LUT). In individual channels, white pixels indicate low fluorescence intensity and black pixels indicate higher levels of fluorescence. In merged images using this LUT, black pixels indicate the co-localization of fluorescence displayed in individual magenta and green channels. Confocal image stacks were prepared for publication in Fiji (*Schindelin et al., 2012*) or Photoshop (Adobe).

## Quantification of Merkel cells and hair follicle lengths

Confocal image stacks were imported into Neurolucida (MBF Bioscience, Williston, VT). Hair follicles and touch domes were both marked by K17 staining and distinguished by their axial position. Hair follicles jutted down into deeper axial planes whereas touch domes spread laterally into the superficial axial planes. Hair follicles and touch domes could be further distinguished by a change in the arrangement of K17-positive cells: hair follicles possessed neatly arranged columns of cells with slightly more intense immunofluorescence at the border, whereas embryonic touch-dome keratinocytes formed clusters with irregular borders (*Figure 1–video 1*). The localization of Merkel cells in whole-mount specimens was quantified in each individual optical section rather than in projections. Merkel cells that were nestled between K17-immunoreactive cells in hair follicles were counted as 'follicle'. Merkel cells positioned superficial to budding follicles were counted as 'touch dome'.

Primary follicle lengths were traced linearly in three dimensions by stepping through optical sections and drawing lines along the central axis of the hair follicle. For this analysis, we quantified primary follicles that were associated with at least one K8-immunoreactive Merkel cell. One image was excluded from analysis because the confocal image stack did not capture the entire length of the follicle.

## Quantification of Keratin-17 and Neurofilament-Heavy immunoreactive areas

Fiji was used to quantify K17 and NFH areas. First, images were converted to a binary format using the Default thresholding method. Then, a region of interest was drawn around each touch dome based on the footprint of K17 immunolabeling. The measure tool was used to calculate Area, Area fraction and Mean gray values for each region of interest. The Area fraction was then converted to a decimal and multiplied by the total area to obtain the area within the region of interest that was covered by white pixels in square microns. Images with brightly fluorescent artifacts (e.g. skin folds, tears in the specimen) that overlapped the touch dome were excluded from analysis.

For $Krt17^{CreERT2}$;R26$^{DTA}$ experiments, embryos were harvested and stored in PBS. After collecting a sample for genotyping, specimens were randomized and blinded by an independent experimenter. Data were de-blinded after all imaging and quantification were complete.

## Quantitative analysis of NFH-positive neuronal populations

For line cross analysis (*Figure 5E*), confocal image stacks were first projected two-dimensionally in ImageJ and then imported into Illustrator (Adobe). The rectangular grid tool was used to overlay a series of lines spaced 3 μm apart, with the '0' line drawn at the skin surface. A modified Sholl analysis was performed to count the number of crossings for either NFH-positive or tdTomato-positive afferent branches. Each marker was quantified independently. A crossing was defined as an intersection between an NFH-positive and tdTomato-positive branch and an overlaid line.

The kinetics of Merkel-cell contacts were quantified in $Ntrk3^{tdTomato}$ mice (N = 2–3 mice per stage; TdTomato-positive neurons and NFH-positive neurons) and wildtype mice (N = 2–3 mice per stage; NFH-positive neurons). For each touch dome, the total number of K8-positive cells were counted in whole mount and cryosection images. Then, the number of K8-positive cells within one pixel from an immunoreactive afferent branch was counted as contacted by either tdTomato- or NFH-positive innervation. For each developmental stage, data were pooled across all experimental replicates. The percentage of contacted Merkel cells across developmental stages was then plotted and fit with an exponential curve.

To quantify neural complexity measures, axial confocal image stacks were imported into Neurolucida. Afferents projecting toward the epidermis were traced in three dimensions. For each touch-dome arbor, the number of terminal endings, the highest branching order, and the neuronal length were quantified. A neural complexity index was also calculated according to the following equation:

Complexity = (sum of the terminal orders + # terminals) · (total length/# primary branches)

Immunoreactivity of dorsal root ganglion somata was quantified by first using the Default threshold method to convert each image to binary in Fiji. Then, nuclei were identified by the absence of immunofluorescence at the center of each soma. Only somata with visible nuclei were quantified. Somata were designated as 'positive' for each marker by observation of suprathreshold immunofluorescent signal.

## Depth coding

The ImageJ Temporal Color Code tool (*Schneider et al., 2012*) was used to depth code confocal image stacks. A new lookup table was generated inversely to the 'Ice' lookup table and was used to pseudo color each image in the stack based on its axial depth. Images were then projected and inverted to a white background in Photoshop. Since skin thickness changes during development, and each specimen was depth coded over its full thickness, the span of the color code scale for each image varies.

### *In utero* lentiviral injection

Preparation of high-titer lentivirus and *in utero* ultrasound-guided lentiviral injections were performed as previously described (*Beronja et al., 2010*). *Noggin* and *Shh* sequences were cloned by PCR to generate lentiviral Dox-inducible constructs as previously described (*Lu et al., 2016*). *Krt14^rtTA* transgenic males (CD1 strain) were mated with CD1 wild-type females. At E9.5, a subset of embryos in each pregnant dam was injected with high-titer lentivirus. Pregnant dams were fed Dox (2 mg/kg) chow either from E14.5–P0, E17.5–P0 or E16-E18 to activate the rtTA transcription factor. Pups were sacrificed at birth (P0) or P4 for tissue collection. Pups that had successfully taken up virus expressed GFP visibly in their skin, which was assessed by viewing all animals with a fluorescent stereoscope. Tail tissue from all pups was collected for genotyping after sacrificing to determine *Krt14^rtTA* expression. Specimens were then blinded by an independent experimenter. *Krt14^rtTA*-positive GFP-positive mice were analyzed as experimental biological replicates. *Krt14^rtTA*-positive GFP-negative mice and *Krt14^rtTA*-negative GFP-positive mice were analyzed as controls. Data were de-blinded after all imaging and analyses were complete.

### Statistics

Statistical analyses were performed with Prism 5 (Graphpad). For parametric data with three or more groups, one-way ANOVAs were followed by Tukey's *post hoc* analysis for between-group comparisons. Two-way ANOVA was used to compare the effect of two factors and was followed by Sidak's multiple comparisons test. Unpaired, non-parametric data were analyzed using the Kruskal-Wallis test with Dunn's *post hoc* comparison. Student's two-tailed *t* test was used to compare means of two normally distributed groups. Mann Whitney test was used to compare means of non-normally distributed groups. Linear regressions were performed in Prism and Pearson's correlation values were assessed based on these fits. The normality of population data was assessed using the D'Agostino and Pearson omnibus normality test, with $p < 0.05$ indicating non-normality.

## Acknowledgments

We thank Rachel Clary for assistance with blinding data and for her insightful comments on experimental design and analysis. Drs. Richard Behringer and Mary Dickinson assisted with *Bmp4^CFP* mice. Megan Sribour and John Levorse assisted with *in utero* injections. Drs. Ling Bai and David Ginty provided *Ntrk3^tdTomato* skin specimens for immunohistochemistry. Dr. David Owens provided *Krt17^CreERT2* mice. Drs. Carol Mason, Jane Dodd, Wesley Grueber, David Owens, Michael Rendl, Frank Rice and members of the EAL and EF laboratories provided helpful discussions. Imaging and tissue processing core facilities were supported by the Columbia University EpiCURE Center (NIAMS P30AR069632) and the Confocal and Specialized Microscopy Shared Resource of the Herbert Irving Comprehensive Cancer Center (NCI P30CA013696). This research was supported by NINDS R01NS073119 and NIAMS R01AR051219 (to EAL) and NIAMS R01AR050452 (to EF). BAJ was supported by NINDS F31NS094023 and C Lu was supported by NIAMS K01AR066073. EF is an Investigator of the Howard Hughes Medical Institute. EAL is supported by the Thompson Family Foundation Initiative in CIPN and Sensory Neuroscience.

## Additional information

### Competing interests

Elaine Fuchs: Reviewing editor, *eLife*. The other authors declare that no competing interests exist.

### Funding

| Funder | Grant reference number | Author |
|---|---|---|
| National Institute of Neurological Disorders and Stroke | F31NS094023 | Blair A Jenkins |
| National Institute of Arthritis and Musculoskeletal and Skin Diseases | K01AR066073 | Catherine P Lu |

| Howard Hughes Medical Institute | Investigator | Elaine Fuchs |
| National Institute of Arthritis and Musculoskeletal and Skin Diseases | R01AR050452 | Elaine Fuchs |
| National Institute of Neurological Disorders and Stroke | R01NS073119 | Ellen A Lumpkin |
| National Institute of Arthritis and Musculoskeletal and Skin Diseases | R01AR051219 | Ellen A Lumpkin |
| National Cancer Institute | P30CA013696 | Ellen A Lumpkin |
| National Institute of Arthritis and Musculoskeletal and Skin Diseases | P30AR069632 | Ellen A Lumpkin |
| Thompson Family Foundation | Initiative in CIPN and Sensory Neuroscience | Ellen A Lumpkin |

The funders had no role in study design, data collection and interpretation, or the decision to submit the work for publication.

## Author contributions

Blair A Jenkins, Conceptualization, Data curation, Formal analysis, Funding acquisition, Validation, Investigation, Visualization, Methodology, Writing—original draft, Writing—review and editing, Conceptualization—formulation of overarching research goals; Natalia M Fontecilla, Conceptualization, Resources, Investigation, Writing—review and editing; Catherine P Lu, Resources, Investigation, Methodology, Writing—review and editing, Conceptualization: evolution of research goals; Elaine Fuchs, Conceptualization, Resources, Methodology, Writing—review and editing, Conceptualization: evolution of research goals; Ellen A Lumpkin, Conceptualization, Formal analysis, Supervision, Funding acquisition, Methodology, Writing—original draft, Project administration, Writing—review and editing, Conceptualization: formulation of overarching research goals

## Author ORCIDs

Blair A Jenkins (iD) http://orcid.org/0000-0003-0226-5021
Elaine Fuchs (iD) https://orcid.org/0000-0002-0978-5137
Ellen A Lumpkin (iD) http://orcid.org/0000-0002-1166-3374

## Ethics

Animal experimentation: This study was performed in accordance with the recommendations in the Guide for the Care and Use of Laboratory Animals of the National Institutes of Health. Animal studies were approved by the Institutional Animal Care and Use Committees of Columbia University (Protocol #AC-AAAV6459 and AC-AAAM1053) and The Rockefeller University (Protocol #16874).

## Decision letter and Author response

Decision letter https://doi.org/10.7554/eLife.42633.024
Author response https://doi.org/10.7554/eLife.42633.025

# Additional files

## Supplementary files

• Transparent reporting form
DOI: https://doi.org/10.7554/eLife.42633.022

## Data availability

No major datasets (e.g., sequence data) were generated through this study. All data generated or analyzed during this study are included in the manuscript.

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
