## [Decision Letter]

Thank you for submitting your manuscript "The cellular basis of mechanosensory Merkel-cell innervation during development" (2018-42633) to *eLife*. Three experts reviewed your manuscript, and their assessments, together with my own, form the basis of this letter. As you will see, all of the reviewers were impressed with the importance and novelty of your work.

I am including the three reviews at the end of this letter, as there are a variety of specific and useful suggestions in them. Please note especially reviewer #3's revision suggestions 1-3. We appreciate that the reviewers' comments cover a broad range of suggestions for improving the manuscript. We look forward to receiving your revised manuscript.

*Reviewer #1:*

This is a very interesting manuscript that explores the early development of touch domes and associated Merkel cells and sensory afferent terminals in mouse skin. The study defines the time windows within which touch dome precursors appear and produce Merkel cells and become associated with sensory terminals; identify an apparently discrete population of BMP expressing mesenchymal cells that contribute to touch dome development; identify a previously unrecognized population of TrkC negative NFH positive sensory terminals that transiently innervate the touch dome with a pattern distinct from that of those neurons associated more intimately with Merkel cells; establish the necessity and sufficiency of K17 expressing epithelial cells for touch dome innervation; and build upon previous studies that demonstrate the dispensibility of Merkel cells for the recruitment of sensory terminals to the vicinity of touch domes. The data are clearly presented, and the experiments appear to be well designed, rigorously analyzed, and thoughtfully discussed. This work represents a significant contribution the fields of skin development and sensory biology and should be of interest to a broad audience.

*Reviewer #2:*

In this interesting paper, Jenkins et al., showed that touch dome patches emerge at E16.5 and are in close proximity with the BMP4-expressing dermal fibroblasts (needs to be confirmed). Overexpressing noggin (to suppress BMP4 signaling) in epithelial cells reduces K17-expressing keratinocytes (shown to be progenitors of Merkel cells in previous publications, PMID:23727240), Merkel cells, and touch dome innervating sensory nerve endings. Overexpressing shh in K17-expressing keratinocytes expands k17-positive population and touch dome innervation by NFH-positive sensory nerve endings without changing the number of Merkel cells, suggesting that touch tome innervation is dictated by K17-positive keratinocytes rather than Merkel cells.

1) The identity of the BMP4-expressing dermal fibroblasts has not been determined although it has been mentioned in the abstract. Since epithelial overexpression of noggin will likely affect larger populations of skin cells than the BMP4-expressing cells, the specificity of inhibition of BMP4 signaling by noggin overexpression is not clear. This should be discussed. Is there a better way to selectively manipulate the BMP4-expressing cells in the skin?

2) Although ablation of K17-expressing keratinocytes reduced the number of Merkel cells, overexpressing shh in K17-expressing keratinocytes expanded the K17 population and increased NFH-positive skin innervation but did not affect the number of Merkel cells. Since Owens lab showed that Merkel cells are likely derived from the K17-positive progenitors, it is not understood why the number of Merkel cells was not also increased. Is it caused by compromised differentiation capability of the expanded K17-expressing keratinocytes?

*Reviewer #3:*

Overall, this is an interesting and well-written manuscript describing some of the mechanisms underlying both the formation and initial innervation of the touch dome mechanoreceptor. The authors use multiple innovative state of the art approaches, generally the data collection appears rigorous, and analyses are appropriate. I recommend the following improvements to the manuscript.

1) The manuscript employs multiple different experiments which are not tied together as well as they could be. A summary illustration or graphical abstract describing the process of touch dome development, with an emphasis on findings from the manuscript, might improve the impact of the research.

2) It was unclear to me from the images that the touch dome was hyper-innervated at E18.5 followed by a retraction during postnatal development. Some quantification at these two ages would clarify this finding. This could be accomplished from images already collected, if enough were collected to justify the conclusions.

3) The experiment testing whether ectopic K17 cells are sufficient to encourage innervation seems important. However, it is relegated to a single supplementary image. Presumably this is because these data were not quantified. However, it could be worth the effort to quantify these results and move them into the main body of the manuscript.

---

## [Author Response]

Reviewer #2:

In this interesting paper, Jenkins et al., showed that touch dome patches emerge at E16.5 and are in close proximity with the BMP4-expressing dermal fibroblasts (needs to be confirmed). Overexpressing noggin (to suppress BMP4 signaling) in epithelial cells reduces K17-expressing keratinocytes (shown to be progenitors of Merkel cells in previous publications, PMID:23727240), Merkel cells, and touch dome innervating sensory nerve endings. Overexpressing shh in K17-expressing keratinocytes expands k17-positive population and touch dome innervation by NFH-positive sensory nerve endings without changing the number of Merkel cells, suggesting that touch tome innervation is dictated by K17-positive keratinocytes rather than Merkel cells.1) The identity of the BMP4-expressing dermal fibroblasts has not been determined although it has been mentioned in the abstract. Since epithelial overexpression of noggin will likely affect larger populations of skin cells than the BMP4-expressing cells, the specificity of inhibition of BMP4 signaling by noggin overexpression is not clear. This should be discussed. Is there a better way to selectively manipulate the BMP4-expressing cells in the skin?

Aside from BMP4, the molecular identity of touch-dome dermal cells remains unknown; therefore, we have not yet devised a better way to selectively manipulate the touch-dome BMP4-expressing dermal cells. We recognize that noggin is likely to have myriad effects on morphogen signaling in skin, including signaling from BMPs other than BMP4. Thus, Noggin will not influence solely the novel cell population that we have identified. We have discussed this limitation in the subsection “Complex signaling cascades give rise to developing Merkel cells”.

*2) Although ablation of K17-expressing keratinocytes reduced the number of Merkel cells, overexpressing shh in K17-expressing keratinocytes expanded the K17 population and increased NFH-positive skin innervation but did not affect the number of Merkel cells. Since Owens lab showed that Merkel cells are likely derived from the K17-positive progenitors, it is not understood why the number of Merkel cells was not also increased. Is it caused by compromised differentiation capability of the expanded K17-expressing keratinocytes?*

Ectopic K17-expressing keratinocytes induced early (E12.5) are capable of producing Merkel cells (Nguyen et al., 2018; Perdigoto et al., 2016). Our data indicate that K17-expressing keratinocytes induced at late embryonic stages lack potency to give rise to Merkel cells, perhaps because they lack positional cues from hair follicles. This dichotomy is noted in the Results section. Moreover, Wright et al., (2015) previously proposed the existence of two populations of K17-expressing cells in the adult touch dome: Atoh1+/K17+ cells, which give rise to Merkel cells, and Atoh1−/K17+ cells, which do not.

Reviewer #3:

Overall, this is an interesting and well-written manuscript describing some of the mechanisms underlying both the formation and initial innervation of the touch dome mechanoreceptor. The authors use multiple innovative state of the art approaches, generally the data collection appears rigorous, and analyses are appropriate. I recommend the following improvements to the manuscript.1) The manuscript employs multiple different experiments which are not tied together as well as they could be. A summary illustration or graphical abstract describing the process of touch dome development, with an emphasis on findings from the manuscript, might improve the impact of the research.

Thank you for this suggestion. A summary is presented in a new Figure 8.

2) It was unclear to me from the images that the touch dome was hyper-innervated at E18.5 followed by a retraction during postnatal development. Some quantification at these two ages would clarify this finding. This could be accomplished from images already collected, if enough were collected to justify the conclusions.

The line crossing analysis of E18.5 touch domes was performed on 50 touch domes from three mice. We did not collect a comparable data set from postnatal animals because published reconstructions demonstrate that NFH-positive afferents terminate predominantly at the Merkel-cell layer in adult skin (Lesniak et al., 2014; Marshall et al., 2016). To bolster this conclusion, we have included a supplemental figure showing the remainder of our dataset of P21 touch domes imaged in cryosections (Figure 5—figure supplement 2). In agreement with published results from adult skin, we did not observe NFH immunoreactivity extending beyond the Merkel-cell layer in P21 touch domes (N=14 touch domes from two mice).

3) The experiment testing whether ectopic K17 cells are sufficient to encourage innervation seems important. However, it is relegated to a single supplementary image. Presumably this is because these data were not quantified. However, it could be worth the effort to quantify these results and move them into the main body of the manuscript.

We included these data in a supplemental figure because the time point of tissue collection for this experiment (P4) did not match the time points we used for quantifying innervation in main text studies (E19.5–P0). These data were collected from tissue blocks fortuitously obtained in a previous study of epidermal appendage specification (Lu et al., 2016). For this manuscript, we sectioned existing tissue blocks and stained for Merkel cells and NFH^+^ afferents. We were excited by the enhancement of innervation in areas with K17-cell expansion and felt that including these qualitative data in the manuscript would strengthenour conclusions.

Obtaining a well-powered dataset at P0 for quantification would require producing more TRE-Shh virus and repeating the *in utero* targeting experiments. Unfortunately, we cannot perform this analysis within *eLife’s* revision timeframe. We hope the editors and reviewers agree that the qualitative data are a valuable addition to the study in supplemental materials. To further bolster the qualitative results, we have included examples of additional fields of view in the revised manuscript.